# Molecular Classification to Prognosticate Response in Medically Managed Endometrial Cancers and Endometrial Intraepithelial Neoplasia

**DOI:** 10.3390/cancers13112847

**Published:** 2021-06-07

**Authors:** Allison M. Puechl, Daniel Spinosa, Andrew Berchuck, Angeles Alvarez Secord, Kerry E. Drury, Gloria Broadwater, Janice Wong, Regina Whitaker, Nicolas Devos, David L. Corcoran, Kyle C. Strickland, Rebecca A. Previs

**Affiliations:** 1Atrium Health, Division of Gynecologic Oncology, Levine Cancer Institute, Charlotte, NC 29204, USA; 2Department of Obstetrics & Gynecology, Duke University Medical Center, Durham, NC 27710, USA; daniel.spinosa@duke.edu (D.S.); Kerry.drury@duke.edu (K.E.D.); janice.wong@duke.edu (J.W.); 3Duke Cancer Institute, Duke University Medical Center, Durham, NC 27710, USA; andrew.berchuck@duke.edu (A.B.); angeles.secord@duke.edu (A.A.S.); regina.whitaker@duke.edu (R.W.); rebecca.previs@duke.edu (R.A.P.); 4Duke Cancer Institute Biostatistics, Durham, NC 27710, USA; gloria.broadwater@duke.edu; 5GCB, Department of Biostatistics & Bioinformatics, Duke University, Durham, NC 27710, USA; Nicolas.devos@duke.edu; 6Duke Center for Genomics and Computational Biology, Durham, NC 27710, USA; David.corcoran@duke.edu; 7Department of Pathology, Duke University Medical Center, Durham, NC 27710, USA; kyle.strickland@duke.edu

**Keywords:** uterine cancer, endometrial cancer, levonorgestrel intrauterine device, molecular classification, mismatch repair deficiency, *POLE* mutation

## Abstract

**Simple Summary:**

Uterine cancer is the most common gynecologic cancer. The treatment for women with a newly diagnosed uterine cancer is surgery to remove the uterus, fallopian tubes, ovaries, and lymph nodes. For women who desire future pregnancy or are not healthy enough to undergo surgery with hysterectomy, treatment with hormonal therapy, using the levonorgestrel intrauterine device (IUD), is an option. Response rates to this type of therapy are promising and range between 50% and 88%. This project explores a molecular classification scheme and applies it to women with uterine cancer and pre-cancer who opt for treatment with an IUD. Evaluating the genetic alterations underlying tumor development can identify patients who would be more or less likely to respond to treatment with IUDs. Additionally, this knowledge can help physicians counsel women with uterine cancer who may be interested in non-surgical options about her odds of having her cancer respond to a hormonal treatment.

**Abstract:**

Background: The aim of this study was to evaluate whether molecular classification prognosticates treatment response in women with endometrial cancers and endometrial intraepithelial neoplasia (EIN) treated with levonorgestrel intrauterine system (LNG-IUS). Methods: Patients treated with LNG-IUS for endometrial cancer or EIN from 2013 to 2018 were evaluated. Using immunohistochemistry and single gene sequencing of *POLE*, patients were classified into four groups as per the Proactive Molecular Risk Classifier for Endometrial cancer (ProMisE): *POLE*-mutated, mismatch repair-deficient (MMRd), p53 wild type (p53wt), and p53-abnormal (p53abn). Groups were assessed relative to the primary outcome of progression or receipt of definitive treatment. Results: Fifty-eight subjects with endometrioid endometrial cancer or EIN treated with LNG-IUS were included. Of these, 22 subjects (37.9%) had endometrial cancer and 36 subjects (62.1%) had EIN. Per the ProMisE algorithm, 44 patients (75.9%) were classified as p53wt, 6 (10.3%) as MMRd, 4 (6.9%) as p53abn, and 4 (6.9%) as *POLE*-mutated. Of the 58 patients, 11 (19.0%) progressed or opted for definitive therapy. Median time to progression or definitive therapy was 7.5 months, with p53abn tumors having the shortest time to progression or definitive therapy. Conclusions: Molecular classification of endometrial cancer and EIN prior to management with LNG-IUS is feasible and may predict patients at risk of progression.

## 1. Introduction

The majority of women with endometrial cancer or endometrial intraepithelial neoplasia (EIN) undergo surgical management [1]. Patients who desire fertility or are not surgical candidates due to medical co-morbidities may opt for medical management with progestins, including oral agents (medroxyprogesterone acetate or megestrol acetate) or the levonorgestrel intrauterine system (LNG-IUS). Data comparing the efficacy of oral progestins versus the LNG-IUS are limited, but at least one study reported higher rates of regression with LNG-IUS over oral progestins in endometrial hyperplasia [2]. Multiple studies have reported successful treatment of EIN, hyperplasia, or endometrial cancer with the LNG-IUS alone or in combination with oral progestins with variable complete response rates depending on the diagnosis [3,4,5,6,7,8,9]. Most recently, a phase II trial of LNG-IUS treatment in early endometrial cancer and EIN identified a 12-month response rate of 47.6% for endometrial cancer and 80.6% for EIN [10]. Additionally, four (9.5%) patients relapsed after initial response: one patient had hyperplasia, and three patients had endometrioid endometrial cancer [10]. Similarly, another randomized control trial found a pathological compete response rate with LNG-IUS of 43% for endometrial cancer and 82% for EIN, as well as evidence that weight loss improved this response [11].

Previous work has identified clinical and radiologic factors that may be associated with favorable responses, including: thickness of the endometrial stripe on imaging, a younger age at diagnosis, history of prior pregnancy, and shorter menstrual cycles. The role of body mass index as a predictor is less well understood [12,13,14,15]. Previous work has identified biomarkers that predict initial progesterone resistance and include proliferation markers (Ki67), estrogen-regulated genes, and genes within the *Wnt* signaling pathway. Further studies are required to determine the mechanism of progesterone resistance given the complex functions and isoforms of the progesterone receptor [10]. Additionally, endometrial cancers with abnormal mismatch repair immunohistochemistry staining patterns and weak stromal expression of progesterone receptor beta are more likely to have resistance to hormonal treatment [16,17,18,19].

Historically, endometrioid cancers have been classified based on histology, with well-differentiated cancers designated as type 1 and high-grade cancers as type 2. This schema has been replaced by a classification of underlying genomic alterations. The Cancer Genome Atlas (TCGA) identified four molecular subtypes of endometrial cancer: *POLE*-mutated, microsatellite unstable, copy number-low, and copy number-high [20]. This system predicts progression-free survival, with *POLE*-mutated tumors associated with the best outcomes whereas copy number-high tumors demonstrated the worst prognosis [20].

TCGA classification requires genomic sequencing, which is not practical or cost effective for routine clinical practice. The “Proactive Molecular Risk Classifier for Endometrial cancer” (ProMisE) was developed as an alternative classification system [21,22,23]. ProMisE approximates TCGA categorizations, including survival curves, using single gene sequencing of *POLE* and protein expression analysis via immunohistochemistry, and it is a clinically feasible method to identify TCGA groups [21,22,23]. To date, molecular classification has been applied to endometrial cancer patients receiving standard-of-care therapy with hysterectomy and used to describe the frequency of molecular alterations in hormonally treated endometrial cancers [24,25]. This molecular classification has not been previously applied to patients with EIN. EIN is a monoclonal proliferation of endometrial glands and a known pre-cursor to endometrial adenocarcinoma, some of which have underlying abnormal P53 expression via immunohistochemistry [26]. The significance of abnormal *TP53* expression in a pre-cursor lesion like EIN is somewhat unclear, but this finding is an accurate surrogate for *TP53* mutations in endometrial cancer [27,28,29]. Ultimately, about 40% of patients with EIN receive a diagnosis of adenocarcinoma on final pathology after hysterectomy [30].

There is a risk of disease progression when managing patients with EIN and endometrial cancer using the LNG-IUS because the uterus remains in situ [31]. Currently, serial biopsies and imaging are the only methods we have available to monitor patients for progression.

The objective of this study was to determine if molecular classification, via ProMisE, prognosticated response in women with endometrial cancer or EIN treated with an LNG-IUS. Our secondary objective was to validate whether molecularly classifying endometrial tumors using ProMisE could be performed on the specimens obtained for initial diagnosis in women receiving LNG-IUS for cancer or EIN.

## 2. Materials and Methods

The Duke Institution Review Board (IRB# Pro00091003) approved this retrospective, single-institutional study. Patients with a histologic diagnosis of endometrioid endometrial adenocarcinoma or endometrial intraepithelial neoplasia (EIN) receiving treatment with a levonorgestrel intrauterine system (LNG-IUS) between January 2013 and January 2018 and had sufficient tissue evaluable for analysis were included. Patients were treated with the LNG-IUS (obtained from pharmacy records) at the discretion of the attending physician for any indication, including, but not limited to, desire for fertility preservation, comorbid medical conditions that precluded definitive treatment with surgery or radiation, and/or patient preference. Patients were included even if they received hormonal therapy prior to or concurrently with the LNG-IUS. Pathologic diagnoses were obtained via endometrial biopsy or curettage. Patients were required to have at least 1 follow-up biopsy after 3 months of treatment. Excluded patients were those who did not have archived tissue available at our institution and had no follow-up biopsies available. In addition to demographic data, information obtained from the medical record included: date of diagnosis, age, method of sampling (dilation and curettage or endometrial biopsy), histology, date of LNG-IUS placement, indication for LNG-IUS placement, pertinent imaging findings, dates and histology of repeat endometrial biopsy(s), whether a change in treatment modality occurred, date of treatment change, and date of last contact.

The primary outcome was progression or non-response with definitive therapy for each molecular group. The secondary outcome of the study was to evaluate whether molecular classification using the ProMisE algorithm could be performed on the specimens obtained for initial diagnosis via endometrial biopsy or curettage.

Repeat endometrial biopsies for all subjects were characterized as 1 of the following: (1) benign complete response, (2) atypical—(hormonal effect) partial response, (3) atypical—(improved disease) partial response, (4) atypical—(no better/no worse) stable disease, (5) atypical—progressive disease, or (6) insufficient tissue for evaluation. “Benign compete response” was defined as a previously abnormal endometrial specimen that demonstrated normal endometrial tissue on subsequent biopsy. “Atypical—(hormonal effect) partial response” was defined by a repeat biopsy demonstrating histologic progestin effect. “Atypical—(improved disease) partial response” was defined as a repeat biopsy with less severe, but not normal, histology (i.e., grade 1 endometrioid endometrial cancer on initial biopsy with EIN or atypical endometrial cells on repeat biopsy). “Atypical—(no better/no worse) stable disease” was defined as no change in the abnormal histology between biopsies. “Atypical—progressive disease” was defined as an increase to a more severe histology on subsequent biopsy (i.e., EIN biopsy followed by grade 1 endometrioid endometrial cancer biopsy, grade 1 endometrioid endometrial cancer biopsy followed by grade 2 endometrioid EC biopsy, etc.). “Definitive therapy” was defined as a change in treatment modality from LNG-IUS to radiation, chemotherapy, or surgery.

### 2.1. Application of ProMisE Algorithm

Archival tissue from the initial endometrial biopsy was evaluated according to the ProMisE algorithm (Figure 1) [21,22,23]. Tumors were categorized into 1 of the 4 ProMisE molecular categories: *POLE*-mutated, mismatch repair-deficient (MMRd), p53 wild type (p53wt), or p53-abnormal (p53abn). In the first step of the algorithm, formalin-fixed paraffin embedded (FFPE) tissues were evaluated via immunohistochemistry (IHC) for expression of the mismatch repair (MMR) proteins MLH1, MSH2, MSH6, and PMS2. Sections were deparaffinized in xylene and declining grades of ethanol prior to rehydration. After antigen retrieval with citrate buffer pH 6.0, (Sigma-Aldrich, St. Louis, MO, USA), the sections were blocked with 3% hydrogen peroxide and protein blocking solution (Background Terminator, BioCare Medical, Pacheco, CA, USA) at room temperature. The sections were incubated with the following primary antibodies overnight at 4 °C: MLH1 (1:100 dilution, clone G168-728, Sigma-Aldrich, St. Louis, MO, USA), MSH2 (1:1000 dilution, clone G2019-1129, Sigma-Aldrich, St. Louis, MO, USA), PMS2 (1:100 dilution, clone MRQ-28, Sigma-Aldrich, St. Louis, MO, USA), and MSH6 (1:200 dilution, clone 44, Sigma Aldrich, St. Louis, MO, USA). After washing the slides with tris buffered saline, protein expression was visualized using the 4 Plus Universal Detection system (BioCare Medical, Pacheco, CA, USA) for 10 min at room temperature. Slides were developed with 3, 3”-diaminobenzidine chromogen (Vector Laboratories, Burlingsame, CA, USA) and counterstained using a modified Lillie-Mayer Hematoxylin (BioCare Medical, Pacheco, CA, USA).

An anatomic pathologist with experience in gynecologic pathology (KS) evaluated MMR immunohistochemistry. If tumors demonstrated abnormal expression (loss) of MMR protein nuclear expression by IHC, the tumor was classified into the mismatch repair-deficient group (Figure 1). Protein expression in normal endometrial glands and stroma served as an internal positive control.

Genomic DNA (gDNA) was isolated from FFPE using the GeneRead DNA FFPE kit (Qiagen, Germantown, MD, USA), and digital DNA sequencing of the *POLE* exonuclease domain (residues 268–471) was performed using a QIAseq Targeted DNA Panels kit from Qiagen. A custom targeted panel was designed to target 3 regions of *POLE*, which are known to be hotspots for pathogenic mutations (exons 9, 13, and 14). Sequencing libraries were prepared from the FFPE-extracted gDNA using the QIAseq Targeted DNA Panels Kit (QIAGEN, Germantown, MD, USA) following the manufacturer’s protocol. Briefly, 100 ng of gDNA was fragmented, end-repaired, and A-tailed. Adapters with sample indexes and UMIs were ligated to the fragmented DNA. Ligated DNA was then subjected to target enrichment by performing an 8-cycle multiplex PCR with the custom-designed QIAseq Targeted DNA Panel primers (QIAGEN, Germantown, MD, USA). After enrichment, the DNA fragments were further amplified using universal primers. The enriched libraries were quantified using Qubit (ThermoFisher Scientific, Waltham, MA, USA) and multiplex. The final library pool was sequenced at 150 bp paired-end on a MiSeq sequencer (Illumina). If a canonical *POLE* mutation was detected, the tumor was classified into the *POLE*-mutated group (Figure 1).

Finally, IHC was performed to evaluate p53 expression. The sections were incubated with p53 primary antibody (clone DO-7, Leica Biosystems, Buffalo Grove, IL, USA). Null or missense mutations for *TP53* are surrogates for the copy number-high TCGA group. Cases with abnormal p53 expression, defined as strong and diffuse expression of p53 in tumor nuclei, were sequenced using a PCR amplification assay followed by conventional Sanger sequencing to detect mutations in the *TP53* tumor suppressor gene. Briefly, the coding regions and intron/exon junctions of *TP53* exons 4 through 11 were amplified from purified genomic DNA by PCR. The primers used for PCR contain M13 universal primer “tails” at their 5′ ends and have 3′ ends that are homologous to their genomic target sequence. PCR products were treated with an exonuclease/phosphatase mixture (ExoSAP-IT) and sequenced using universal M13 forward and reverse primers with the Big Dye Terminator v3.1 Cycle Sequencing Kit (Thermo Fisher Scientific, Walthman, Massachusetts, USA). These products were purified with the Big Dye XTerminator Purification Kit (Thermo Fisher Scientific, Walthman, Massachusetts, USA) and resolved using the ABI Genetic Analyzer. Data were analyzed by the ABI Data Collection software, Sequencing Analysis software, and SeqScape software. Tumors with abnormal expression of p53 constituted the p53abn group. Tumors with wild-type expression of p53 retained expression of mismatch repair proteins, and no evidence of a *POLE* mutation constituted the p53wt group (Figure 1). To ensure accurate classification, all steps of the algorithm were performed for each tumor.

### 2.2. Statistical Analysis

Data were collected retrospectively through chart review and maintained using REDCap software (Research Electronic Data Capture; Vanderbilt University, TN, USA). Summary statistics are provided. Normally distributed continuous variables (age and body mass index (BMI)) were compared using Student’s *t*-test; nonparametric continuous factors were compared using the Wilcoxon rank-sum test and proportions were compared using a chi-squared test. Analyses were conducted using SAS software (Version 9.4; SAS Institute Inc., Cary, NC, USA).

## 3. Results

### 3.1. Patient Characteristics

A total of 64 subjects with endometrioid endometrial adenocarcinoma or EIN were identified. Five subjects had their initial endometrial biopsy specimen obtained at an outside facility and did not have a tissue block available for processing. One subject had insufficient tissue remaining for testing (Figure 1). Fifty-eight (98.3%) had sufficient remaining tissue to undergo algorithm testing.

Demographics and clinico-pathologic features of these patients are detailed in Table 1. The median age of the cohort was 56.4 years (range 24.3–91.1) and median body mass index (BMI) was 46.4 kg/meter^2^ (range 19.9–74.4). In total, 22 subjects (37.9%) and 36 subjects (62.1%) had an initial diagnosis of endometrioid adenocarcinoma and EIN, respectively. With regard to the initial specimens, 50% (29/58) were obtained via endometrial biopsy and 50% (29/58) via dilation and curettage (D&C). Indications for LNG-IUS placement included: 56.9% (33/58) for medical comorbidities precluding surgery, 24.1% (14/58) for fertility-sparing indications, 13.8% (8/58) for patient preference, and 5.2% (3/58) had an LNG-IUS placed at the time of D&C, prior to confirmation of initial histology. In total, 13 (22.4%) patients had prior hormonal therapy prior to LNG-IUS insertion, 12 patients had a p53wt tumor, and 1 patient had a *POLE*-mutated tumor. In total, 6 patients were prescribed medroxyprogesterone acetate prior to LNG-IUS insertion, 6 patients received megestrol acetate, and 1 patient received a leuprolide acetate injection. The average time of hormonal treatment prior to LNG-IUS insertion was 7.9 months (range: 0.5–23.7). Six patients had a partial response to hormonal management and six patients had stable disease prior to LNG-IUS insertion. One patient with a p53wt tumor had progression after 4.4 months of medroxyprogesterone acetate prior to LNG-IUS therapy. The majority of patients (74.1%) had imaging prior to LNG-IUS insertion.

### 3.2. Molecular Classification

Molecular classification via the ProMisE algorithm was successfully applied to all 58 subjects with evaluable endometrial tissue. The distribution of specimens into the four molecular classes was as follows: 75.9% (44/58) p53wt, 10.3% (6/58) MMRd, 6.9% (4/58) *POLE*-mutated, and 6.9% (4/58) p53abn. Table 1 demonstrates patient demographics and clinico-pathologic characteristics stratified by molecular classification. For specimens with an initial histologic diagnosis of adenocarcinoma (*n* = 22), the distribution of specimens into molecular classes was as follows: 77.3% (17/22) p53wt, 18.2% (4/22) MMRd, 4.6% (1/22) p53abn, and 0% *POLE*-mutated. Comparatively, for specimens with an initial histologic diagnosis of EIN (*n* = 36), the distribution of specimens into molecular classes was as follows: 75.0% (27/36) p53wt, 5.6% (2/36) MMRd, 8.3% (3/36) p53abn, and 11.1% (4/36) *POLE*-mutated (Table 1). Examples of IHC staining patterns for p53 and the mismatch repair proteins are depicted in Appendix A.

### 3.3. POLE Mutations

Four tumors, all from patients with an initial biopsy histology of EIN, harbored pathogenic *POLE* mutations, which included P286R (*n* = 3) and V411L (*n* = 1) amino acid substitutions (Table 2).

### 3.4. p53 Immunohistochemistry

In total, 4 of 58 subjects had abnormal (i.e., strong and diffuse) p53 expression by IHC (Figure 2), which was present in patients with EIN (*n* = 3) and low-grade endometrioid endometrial adenocarcinoma (*n* = 1). We did not observe cases with complete absence of p53 expression (i.e., null phenotype) in our cohort. Sequencing demonstrated that all four cases with abnormal p53 IHC harbored nucleotide substitutions in the *TP53* gene, with amino acid substitutions occurring in three cases (Table 3).

### 3.5. Mismatch Repair Immunohistochemistry

In total, 5 of the 58 (10.3%) specimens demonstrated mismatch repair deficiency by immunohistochemistry, 4 of which had initial biopsies of adenocarcinoma and 2 of which had initial biopsies of EIN (Appendix A). Hypermethylation testing was not performed on these specimens, as it would not have resulted in changing the molecular categorization from MMRd.

### 3.6. Treatment Response

The median duration of follow-up for patients was 28.8 months (range: 3.3–141.5) (Appendix A). In total, 97 total biopsies were evaluated from 58 patients, with 43 biopsies consistent with a benign complete response. The median time to first biopsy was 4.4 months (2.2—73.7). Overall, 11 of the 58 subjects (19.0%) undergoing medical management with a LNG-IUS developed histologic progression or opted for definitive therapy in the setting of non-response (Table 4 and Table 5). For the entire study population, the median time to progression or definite therapy was 7.5 months (range: 2.5–40.0). Patients with p53abn tumors had the shortest time to progression or definitive therapy: 5.7 months (range: 4.6–6.8). The patients with *POLE*-mutated and MMRd tumors had the longest median time to progression or definitive therapy with 21.4 or 20.9 months, respectively.

In patients with adenocarcinoma, 6 of 22 (27.2%) required a change in treatment modality from the LNG-IUS, which included 23.5% (4/17) of the p53wt adenocarcinoma group, 25% (1/4) of the MMRd adenocarcinoma group, and 100% (1/1) of the p53abn adenocarcinoma group (Table 4).

Five subjects with EIN required change in treatment modality from the LNG-IUS, which included 7.4% (2/27) of the p53wt EIN group, 50% (1/2) of the MMRd EIN group, 33.3% (1/3) of the p53abn EIN group, and 25% (1/4) of the *POLE*-mutated EIN group (Table 5).

Of the six subjects with adenocarcinoma receiving definitive therapy, 100% (6/6) underwent surgical management (Table 4). Of the five subjects with EIN who received definitive therapy, 80% (4/5) underwent surgical management and 20% (1/5) received radiation after progression to cancer (Table 5).

In the p53wt cohorts, 23.5% required a change in treatment modality from LNG-IUS in the adenocarcinoma group, and 7.4% required a change in treatment modality in the EIN group (Figure 3). The median time to progression was 7.8 months (2.5–40.0). For patients with MMRd or *POLE*-mutated tumors, 33.3% and 25% progressed or required definitive therapy, respectively, regardless of histology. Those patients with a p53abn tumor were the most likely to progress or require definitive therapy (50%) after a median of 5.7 months (4.6–6.8) (Appendix A).

## 4. Discussion

Presented here are the outcomes of a molecularly classified cohort of patients with endometrioid endometrial cancer or EIN managed with LNG-IUS at a single academic institution over a 5-year period. In our study, p53abn tumors demonstrated the worst outcomes, with the highest proportion of patients requiring definitive therapy (50%) and the shortest time to necessitating definitive therapy (5.7 months). In patients with MMRd or *POLE*-mutated tumors, 33.3% and 25% progressed or required definitive therapy, respectively, regardless of histology. The majority of patients had tumors classified as p53wt (78.9%). In this subgroup, the median time to progression was 7.8 months (range: 2.5–40.0), and 13.6% of patients with p53wt tumors progressed or received definitive therapy. Ninety-eight percent of initial pathology specimens obtained via endometrial biopsy or dilation and curettage had enough remaining tissue after initial pathologic diagnosis to undergo application of the molecular classification algorithm. All tested tumors were classified successfully into one of the four molecular subgroups. Four patients (6.9%) were classified as p53abn. The histology of these patients was re-reviewed, and *TP53* sequencing performed to validate IHC findings. Three of four p53abn tumors were predicted to harbor pathogenic amino acid substitutions. Prior to LNG-IUS placement, three of four p53abn tumors had a diagnosis of EIN and one with grade 1 endometrioid endometrial adenocarcinoma.

Previously, TCGA observed that 11.6% of the copy number-high endometrial tumors had grade 1 or 2 endometrioid histologies. TCGA copy number-high tumors, which typically harbor pathogenic *TP53* mutations regardless of histology, demonstrated the worst progression-free survival [20]. This small proportion (11.6%) suggests that serous and high grade histologies were driving poor outcomes in this group and that histology alone cannot predict molecular phenotype or behavior [20]. Although patients with EIN and low-grade endometroid cancers are typically classified as type 1 tumors with indolent behavior, our identification of *TP53* abnormalities in these tumors suggest molecular evaluation may predict patients who may be better served with definitive treatment at diagnosis. Although we did not observe any cases with the complete absence of p53 expression (null phenotype), we expect that this would be an additional staining pattern that could be observed in practice and in future studies.

Other studies have also reported that copy number-high/p53abn endometrioid tumors demonstrate increased biologic aggressiveness [31,32,33,34,35]. Cosgrove et al. molecularly classified 982 endometrioid adenocarcinomas from NRG/GOG210 after completion of definitive surgery and adjuvant therapy [32]. In their study, the “copy number-altered” molecular group of endometrioid endometrial adenocarcinomas (loss of heterozygosity [LOH] at one or more markers) demonstrated the worst progression-free survival (HR 2.31, 95% CI 1.53–3.49) and cancer-specific survival (HR 3.95, 95% CI 2.10–7.44). *POLE*-mutated tumors had the best outcomes, but the differences were not statistically significant [32]. Stelloo et al. evaluated the molecular classification of 861 early-stage endometrioid adenocarcinomas from the PORTEC-1 and PORTEC-2 trials, and those classified in the p53abn group were almost five times more likely (HR 4.86; 95% CI (3.098–7.073)) to die from disease compared to those in the “no specific molecular profile” group (analogous to TCGA copy number-low group or our p53wt cohort) [33].

Unlike the Cosgrove study, which evaluated LOH at three microsatellite repeats for molecular classification of copy number-altered tumors, our study used p53 IHC classification and demonstrated that abnormal p53 expression was associated with worse outcomes [32]. It is unclear whether a LOH-based method or a p53 IHC-based method, such as ProMisE [21,22,23], is a better predictor of outcomes. Nevertheless, both methods demonstrate that this small subset of low-grade endometrioid endometrial adenocarcinomas and EIN are associated with increased biologic aggressiveness. Similarly, a systematic review of the histopathological characterization of the ProMisE endometrial cancer groups (*n* = 912) found that 10% of p53abn tumors were of grade 1 or 2 endometrioid histologies [36]. 

Importantly, 24.1% of our patients desired future fertility. In this patient population, evaluation for Lynch syndrome is critical because endometrial cancer is often the presenting diagnosis for this hereditary cancer syndrome. Lynch syndrome is a hereditary cancer disorder in which germline mutations of DNA mismatch repair genes (MLH1, MSH2, MSH6, and PMS2) lead to an increased risk of uterine, ovarian, gastrointestinal and other tumors. Defective DNA mismatch repair proteins are characteristic of Lynch-associated cancers [37]. Patients with Lynch syndrome, who have a diagnosis of uterine cancer, tend to be younger than those without Lynch syndrome [38,39]. However, the mean age of presentation depends on the specific underlying mutation, and women of childbearing age should be counseled regarding options around fertility preservation [40]. If a specimen is classified as MMRd via immunohistochemistry, appropriate work-up and referral for genetic counseling and germline testing should be performed to confirm the diagnosis [41]. All of our cases had evidence of MLH1 and PMS2 loss, which should prompt *MLH1* promoter hypermethylation testing in the clinical setting. In the absence of *MLH1* promoter hypermethylation, a referral for germline testing should be recommended. For patients who are ultimately diagnosed with Lynch syndrome, guidelines for cancer screening should be followed, risk-reducing surgery may be recommended, and other family members at risk should be offered cascade testing [40].

Whether patients with MMRd endometrial tumors respond as well to progesterone therapy remains unclear [42,43,44,45,46]. In a recent study of young patients with MMRd tumors, Zahkour et al. reported that 7% of women less than 55 years old had abnormal IHC for MMR proteins. These patients had a higher incidence of invasive cancer and were less likely to have disease regression with progestin therapy [17]. Future prospective studies in patients with MMRd tumors will help elucidate if other therapeutic strategies in addition to progestin could overcome disease resistance.

Our findings demonstrate that molecular classification using ProMisE can be performed on the endometrial specimens obtained for initial diagnosis via biopsy or curettage. Seminal studies regarding molecular characterization predominantly used hysterectomy specimens for molecular characterization [20,21,22,47]. Furthermore, the validation of ProMisE in hysterectomy specimens demonstrated that molecular characterization is concordant between initial biopsy and hysterectomy specimens [47]. Our study validates the feasibility of using biopsy specimens for molecular characterization from women who receive medical management of her uterine cancer or EIN.

To our knowledge, our study is the largest study to evaluate the molecular classification of tumors in patients receiving LNG-IUS for cancer or EIN. Nevertheless, this study is limited by its retrospective nature, small sample size, and practice patterns from a single institution. Future evaluation is warranted to determine if molecular classification predicts outcomes for patients considering LNG-IUS therapy for endometrial cancer or EIN. Other future areas of study include combinatorial strategies with the LNG-IUS to improve response within the different molecular subgroups.

Additionally, this is the first study to our knowledge to evaluate the molecular subtyping via ProMisE of EIN specimens with regard to disease progression. Given that up to 42% of patients with EIN [48] may ultimately have concurrent cancer at the time of hysterectomy, accurately risk stratifying those who opt to forego hysterectomy may provide valuable prognostic data, particularly due to our findings of *TP53* mutations in EIN specimens. Further investigation is warranted into the mechanism and significance of this underlying mutation in pre-malignant disease such as EIN.

Taken together, the findings presented here demonstrate that molecular classification using ProMisE can be performed on the specimens obtained for initial diagnosis for patients who receive LNG-IUS for cancer or EIN. We identified patients with EIN and endometrioid endometrial cancer within each of the molecular subgroups including MMRd, *POLE*-mutated, p53abnormal, and p53wt. Seventy-five percent of patients whose tumors were classified at p53abnormal had underlying pathogenic *TP53* mutations, which would have been missed with histology alone. Patients with these tumors were more likely to progress or require definitive therapy than patients with tumors in the other molecular subgroups.

## 5. Conclusions

Hormonal management of EIN and low-grade endometrioid endometrial cancer with the levonorgestrel intrauterine system is often considered in patients who desire future fertility or who are poor surgical candidates due to comorbidities. Our study validates the feasibility of molecular classification of specimens obtained for initial diagnosis via biopsy or curettage for patients with cancer and EIN. Presented here are the outcomes of a molecularly classified cohort of patients with endometrioid endometrial cancer or EIN managed with LNG-IUS at a single academic institution over a 5-year period. Patients were identified in each molecular subgroup, including MMR-deficient, *POLE*-mutated, p53abn, and p53wt. The majority of patients had tumors classified as p53wt tumors (78.9%). The median time to progression was 7.8 months (range: 2.5–40.0), and 13.6% of patients with p53wt tumors progressed or received definitive therapy. In patients with MMRd or *POLE*-mutated tumors, 33.3% and 25% progressed or required definitive therapy, respectively, regardless of histology. A small proportion of patients were identified to have an underlying *TP53* mutation. Patients with p53abn tumors demonstrated the worst outcomes in our cohort: 50% of patients progressed or received definitive therapy. These findings suggest that histology alone cannot predict the molecular phenotype or behavior of a tumor. Further research will determine if molecular classification can predict outcomes for patients considering LNG-IUS therapy for endometrial cancer or EIN.

## Figures and Tables

**Figure 1 cancers-13-02847-f001:**
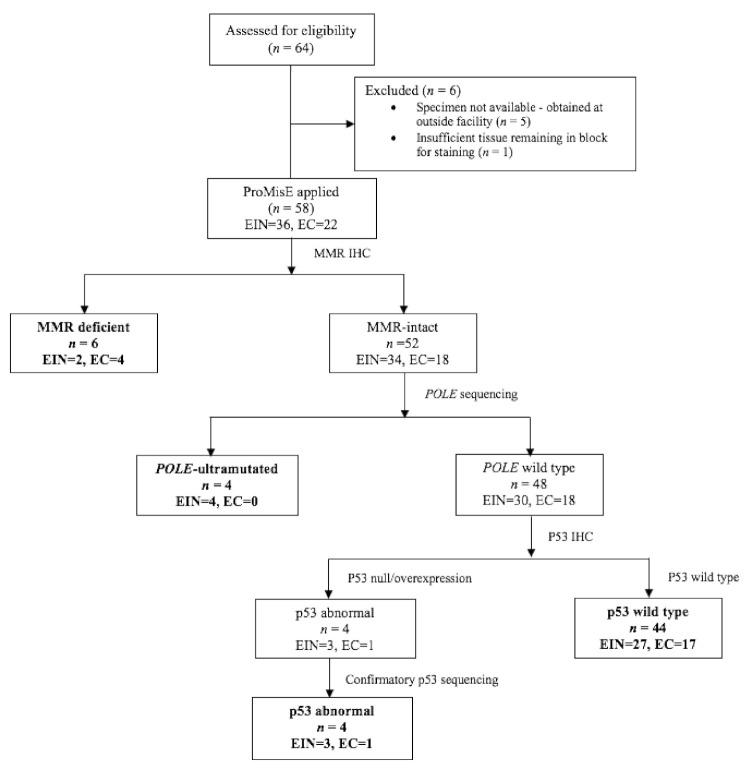
The Proactive Molecular Risk Classifier for Endometrial cancer (ProMisE) molecular classification algorithm [21,22,23]. Our final cohort included 58 tumors with sufficient tissue for testing. Mismatch repair (MMR) status was evaluated by immunohistochemistry targeting of proteins MLH1, PMS2, MSH2, and MSH6, and tumors were categorized as MMR-deficient if they exhibited loss of any of these proteins. MMR-intact tumors were then evaluated for mutations in the exonuclease domain of the polymerase-ε gene (*POLE*) via single gene sequencing. Tumors with wild-type *POLE* sequencing were then subjected to IHC for p53, which identified 4 tumors with strong and diffuse expression of p53. Confirmatory p53 sequencing was performed on these 4 samples to ensure accuracy of IHC. Abbreviations: EC, endometrioid endometrial adenocarcinoma; EIN, endometrial intraepithelial neoplasia.

**Figure 2 cancers-13-02847-f002:**
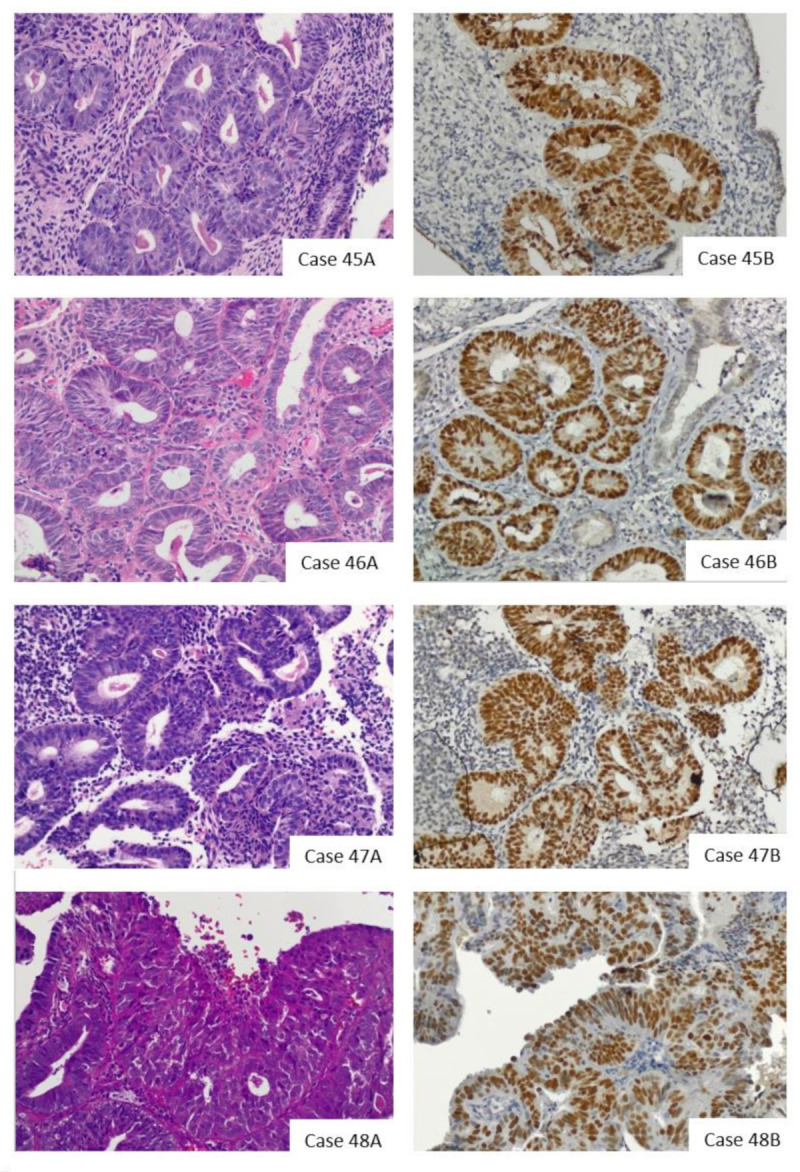
H&E (**A**) and p53 (**B**) staining of the four p53-abnormal (cases 45–48). 45A–48A: H&E images. 45B–48B: IHC demonstrating strong and diffuse p53 staining. Cases 45–47 are endometrial intraepithelial neoplasia (EIN). Case 48 was diagnosed as a grade 1 endometrioid endometrial adenocarcinoma. Photomicrographs (H&E and IHC) were obtained at 20× magnification.

**Figure 3 cancers-13-02847-f003:**
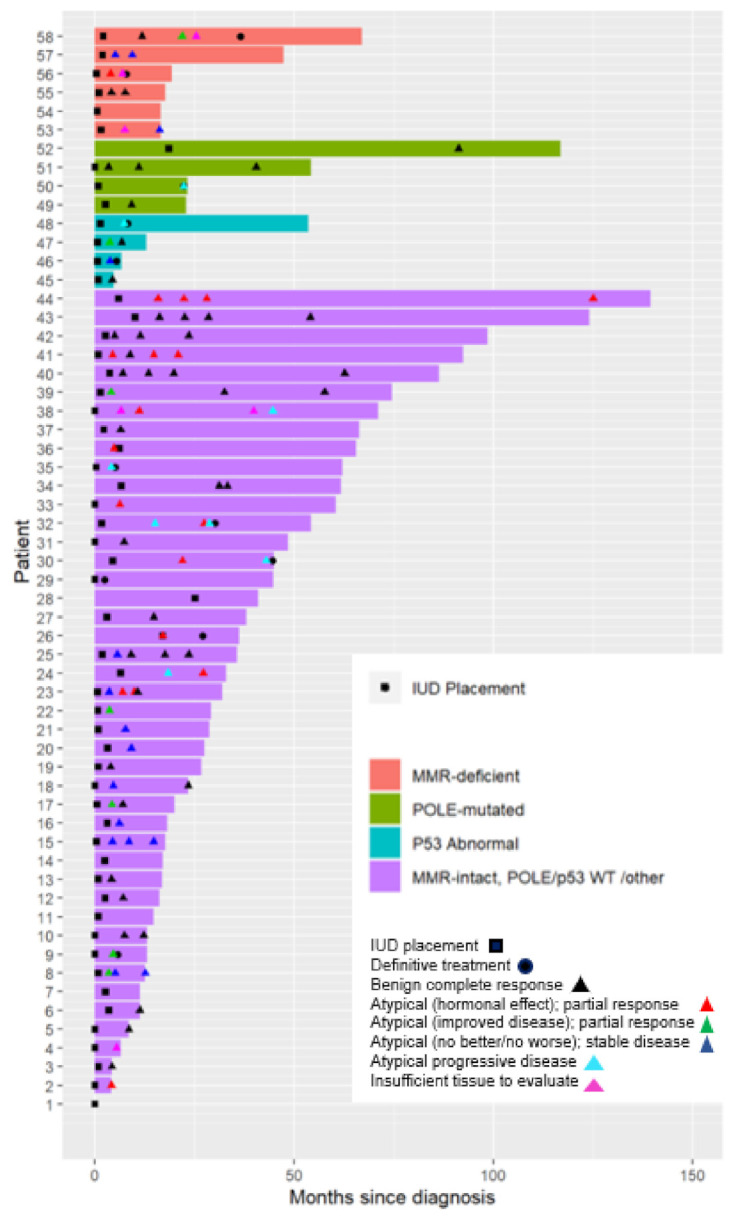
Swimmer plot demonstrating subjects’ (*n* = 58) treatment duration, the time point of diagnosis, LNG-IUS placement, repeat biopsies, and definitive treatment. Abbreviations: MMR: mismatch repair; WT: wild type; IUD: intrauterine device.

**Table 1 cancers-13-02847-t001:** Patient characteristics by molecular classification.

Characteristic	Molecular Classification				
	MMRd (*n* = 6)	*POLE* Mutated (*n* = 4)	p53 Abnormal (*n* = 4)	p53 Wild Type (*n* = 44)	Total (*n* = 58)
**Age at Diagnosis (years)**					
Median (range)	71.7 (51.9–86.8)	55.5 (52.1–68.3)	67.0 (48.1–89.0)	53.1 (24.3–91.1)	56.4 (24.3–91.1)
**Diagnosis**					
Endometrioid cancer	4 (66.7)	0 (0)	1 (25.0)	17 (38.6)	22 (37.9)
Endometrial intraepithelial neoplasia	2 (33.3)	4 (100.0)	3 (75.0)	27 (61.4)	36 (62.1)
**Race**, *n* (%)					
White	4 (66.7)	1 (25.0)	2 (50.0)	25 (56.8)	32 (55.2)
Black	1 (16.7)	3 (75.0)	2 (50.0)	17 (38.6)	23 (39.7)
Asian	1 (16.7)	0 (0)	0 (0)	0 (0)	1 (1.7)
Other	0 (0)	0 (0)	0 (0)	2 (4.6)	2 (3.5)
**Ethnicity**, *n* (%)					
Hispanic or Latino	0 (0)	0 (0)	0 (0)	4 (9.1)	4 (6.9)
Not Hispanic or Latino	6 (100.0)	4 (100.0)	4 (100.0)	40 (90.9)	54 (93.1)
**Parity**					
Median	1.5	0.5	2.5	0.0	1.0
**BMI**, kilograms/meter^2^					
Median (range)	43.0 (21.6–50.3)	46.7 (35.5–53.3)	37.5 (22.1–54.1)	47.2 (19.9–74.4)	46.4 (19.9–74.4)
**ECOG Performance Score**, *n* (%)					
0–1	6 (100.0)	2 (50.0)	4 (100.0)	27 (61.4)	39 (67.2)
2	0 (0)	0 (0)	0 (0)	4 (9.1)	4 (6.9)
3	0 (0)	0 (0)	0 (0)	2 (4.6)	2 (3.5)
Unknown	0 (0)	2 (50.0)	0 (0)	11 (25.0)	13 (22.4)
**History of Other Cancer**, *n* (%)					
No	5 (83.3)	3 (75.0)	3 (75.0)	38 (86.4)	49 (84.5)
Yes	1 (16.7)	1 (25.0)	1 (25.0)	6 (13.6)	9 (15.5)
**Initial specimen obtained through**, *n* (%)					
EMB	1 (16.7)	3 (75.0)	2 (50.0)	23 (52.3)	29 (50.0)
D&C	5 (83.3)	1 (25.0)	2 (50.0)	21 (47.7)	29 (50.0)
**Hormonal therapy prior to LNG-IUS insertion**, *n* (%)					
No	6 (100.0)	3 (75.0)	4 (100.0)	32 (72.7)	45 (77.6)
Yes	0 (0)	1 (25.0)	0 (0)	12 (27.3)	13 (22.4)
**Imaging prior to LNG-IUS****insertion**, *n* (%)					
No	0 (0)	0 (0)	0 (0)	15 (34.1)	15 (25.9)
Yes	6 (100.0)	4 (100.0)	4 (100.0)	29 (65.9)	43 (74.1)
Medical co-morbidities precluding surgery	4 (66.7)	4 (100.0)	3 (75.0)	22 (50.0)	33 (56.9)
Placed at time of D&C	0 (0)	0 (0)	0 (0)	3 (6.8)	3 (5.2)
Patient preference	1 (16.7)	0 (0)	1 (25.0)	6 (13.6)	8 (13.8)

Abbreviations: MMRd: mismatch repair-deficient; BMI: body mass index; ECOG: Eastern Cooperative Oncology Group; EMB: endometrial biopsy; D&C: dilation and curettage; LNG-IUS: levonorgestrel intrauterine system (LNG-IUS).

**Table 2 cancers-13-02847-t002:** Codon and amino acid changes in *POLE*-mutated cases.

Case	Initial Biopsy Histology	Codon Change	Amino Acid Change	Mutation Events	Total Events	Mutation Event Percent
49	EIN	857C > G	Pro286Arg	8	120	6.7
50	EIN	857C > G	Pro286Arg	37	159	23.3
51	EIN	1231G > T	Val411Leu	12	132	9.1
52	EIN	857C > G	Pro286Arg	3	54	5.6

Abbreviations: EIN: endometrial intraepithelial neoplasia; Val: valine; Leu: leucine; Pro: proline; Arg: arginine.

**Table 3 cancers-13-02847-t003:** Nucleotide and amino acid substitutions in tumors with strong and diffuse expression of p53 by immunohistochemistry.

Case	Initial Biopsy Histology	Nucleotide Mutation	Amino Acid Mutation	Mutation Type
45	EIN	c.877G > A	p.Gly293Arg	Amino acid substitution
46	EIN	c.517G > A p.210T > C c.693C > T	p.Val173Met p.Ala70= p.Thr231=	Amino acid substitution Silent mutation Silent mutation
47	EIN	c.651G > A	p.Val217=	Silent mutation
48	Grade 1 EC	c.548C > T c.817C > T	p.Ser183Leu p.Arg273Cys	Amino acid substitution Amino acid substitution

Abbreviations: EC: endometrioid endometrial adenocarcinoma; EIN: endometrial intraepithelial neoplasia: Ser: serine; Leu: leucine; Arg: arginine; Cys: cysteine; Val: valine; Met: methionine; Ala: alanine; Thr: threonine, Gly: glycine.

**Table 4 cancers-13-02847-t004:** Molecular characterization of patients with endometrial adenocarcinoma (*n* = 22) and outcome.

	MMRd (*n* = 4)	*POLE*- Mutated (*n* = 0)	p53 Abnormal (*n* = 1)	p53 Wild Type (*n* = 17)	Total (*n* = 22)
**Progression or required****definitive treatment**, *n* (%)					
No	3 (75.0)	-	0 (0)	13 (76.5)	16 (72.7)
Yes	1 (25.0)	-	1 (100.0)	4 (23.5)	6 (27.3)
**Type of definitive therapy** *n* (%)	*n* = 1		*n* = 1	*n* = 4	*n* = 6
Surgery	1 (100.0)	-	1 (100.0)	4 (100.0)	6 (100.0)
Radiation	0 (0)	-	0 (0)	0 (0)	0 (0.0)
**Final diagnosis** **on hysterectomy specimen** *n* (%)	*n* = 1		*n* = 1	*n* = 4	*n* = 6
No residual disease	0 (0.0)	-	0 (0.0)	1 (25.0)	1 (16.7)
EIN	0 (0.0)	-	0 (0.0)	0 (0.0)	0 (0.0)
Cancer	1 (100.0)	-	1 (100.0)	3 (75.0)	5 (83.33)

Abbreviations: MMRd: mismatch repair-deficient; EIN: endometrial intraepithelial neoplasia.

**Table 5 cancers-13-02847-t005:** Molecular characterization of patients with endometrioid endometrial intraepithelial neoplasia (*n* = 37) and outcome.

Title	MMRd (*n* = 2)	*POLE*- Mutated (*n* = 4)	p53- Abnormal (*n* = 3)	p53 Wild Type (*n* = 27)	Total (*n* = 36)
**Progression or required** **definitive treatment***n* (%)					
No	1 (50.0)	3 (75.0)	2 (66.7)	25 (92.6)	31 (86.1)
Yes	1 (50.0)	1 (25.0)	1 (33.3)	2 (7.4)	5 (13.9)
**Type of definitive therapy**, *n* (%)	*n* = 1	*n* = 1	*n* = 1	*n* = 2	*n* = 5
Surgery	1 (100.0)	0 (0)	1 (100.0)	2 (100.0)	4 (80.0)
Radiation	0 (0.0)	1 (25.0)	0 (0.0)	0 (0.0)	1 (20.0)
**Final diagnosis** **on hysterectomy specimen**, *n* (%)	*n* = 1	*n* = 0	*n* = 1	*n* = 2	*n* = 4
No residual disease	0 (0.0)	-	0 (0.0)	0 (0.0)	0 (0.0)
EIN	0 (0.0)	-	1 (100.0)	0 (0.0)	1 (25.0)
Cancer	1 (100.0)	-	0 (0.0)	2 (100.0)	3 (75.0)

Abbreviations: MMRd: mismatch repair-deficient; EIN: endometrial intraepithelial neoplasia.

## Data Availability

Not applicable.

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
