# Peer review of "Molecular Classification to Prognosticate Response in Medically Managed Endometrial Cancers and Endometrial Intraepithelial Neoplasia"

_cancers, 2021, doi:10.3390/cancers13112847_

Round 1

Reviewer 1 Report

The authors assessed molecular classification to predict the response in conservative treatment of endometrial cancers and endometrial intraepithelial neoplasia. The study is innovative and well-performed. However, the authors did not consider several relevant studies in the field. In particular, several recent studies and systematic reviews and meta-analyses have been published (PMID: 33812697, PMID: 33754186, PMID: 33715893, PMID: 33712277, PMID: 33122144, PMID: 32799597, PMID: 32703491, PMID: 32472441, PMID: 32377987, PMID: 31932106, PMID: 31472940, PMID: 30793281, PMID: 30779338, PMID: 30715091, PMID: 30342311) and should be considered in the Introduction and Discussion section to improve and update the manuscript.

Reviewer 2 Report

  1. Please indicate using the arrows in the figure 2, expression of p53.
  2. Please show the negative control of IHC staining.
  3. PPlease provide percentages of two decimal places, including abstract. 
  4. Please give the name of all regaents used.
  5. The inclussion and exclussion criterias have to be described in more detalis.
  6. How can te authors expalin the ralaticely low number of cases with EIN, EC presented in the figure 1.
  7. Results of IHC staining should be dexribed in more details in the results and discussion section including differences in the density of IHC reaction products and statistucal analysis.
  8. " Prior to LNG-IUS placement, three of four p53abn tumors had a diagnosis of EIN and one with grade 1 endometrioid  dometrial cancer. " Please give the reference
  9. Please detele: "Please add:" form Funding
  10. Number of references and their dates are extremely low and have to be improved.
  11. References should be uptaded.
  12. Please consider change some tables to figures
  13. Graphical abstract is required

Summarazing, the authors should include the rewritten paper, paper with track changes and rebuttal letter which indicating the lines with changing.

Reviewer 3 Report

This is a single-center institutional study involving 58 patients with a histologic diagnosis of endometrial adenocarcinoma or endometrial intraepithelial neoplasia (EIN) receiving treatment with a levonorgestrel intrauterine system (LNG-IUS). The authors here investigated if molecular classification of endometrial cancers and EIN treated with levonorgestrel intrauterine system prognosticates treatment response. As secondary outcome they evaluate whether molecular classification using the ProMisE algorithm could be applied to the initial specimens of endometrial biopsy or curettage. They reported p53abn tumors demonstrated the worst outcomes, in fact these patients required in a large part of cases definitive therapy (50%) and they had the shortest time to progression. The work focuses on a critical topic and findings are interesting. However, there are some flaws that should be addressed. First, the study population is small and it precludes the possibility to draw a conclusion. Research in context is not sufficiently explicated and a more detailed and appropriate background on this topic should be considered. Results section is not easy to be followed and its clarity, in terms of data presentation, should be improved. Finally, a discussion including summary of the findings, interpretation and implication of the results in the clinical practice should be added.

Major points

  1. The research in context seems not to be correctly performed. The introduction lacks of some studies on this topic. Some authors reported the frequency of prognostic molecular alterations within a cohort of endometrial adenocarcinoma women conservatively treated with progestin therapy (Zakhour M et al, BJOG 2017; Falcone et al, European Journal of Obstetrics & Gynecology and Reproductive Biology 2019). Others evaluated the prognostic significance of ProMisE in young women with endometrial cancer (Britton et al, Gynecologic Oncology 2019). Finally, a recent review analyzed the usefulness of immunohistochemical markers in predicting the response to progestogens in endometrial hyperplasia/endometrial cancer. The review reported that an abnormal MMR pattern (including MLH1, MSH2, MSH6, PMS2) strongly predicted poor response (Travaglino A et al, Acta Obstet Gynecol Scand. 2019).
  2. The number of the women included in the study is not clear. In the method section the authors reported that 58 women met the inclusion criteria. In the results section they stated that 64 subjects met the inclusion criteria. Finally, 59 subjects were included to analyze the secondary end point.
  3. In the method section, the authors classified the endometrial biopsies as “benign complete response”, “atypical - (hormonal effect) partial response, atypical - (improved disease) partial response, atypical - (no better/no worse) stable disease, atypical - progressive disease, or insufficient tissue for evaluation.” However, the results have not been shown according to this classification.
  4. In the paragraph about the ProMiSe algorithm, there are expressed too many information about molecular or histopathological techniques and methods for the gene classifications not easy to understand and not useful to the study outcome. These should be made simpler.
  5. Result section is not easy to be followed and the manuscript, in terms of data presentation should be improved.
  6. The authors stated that “the most favorable outcomes were observed in the p53wt cohorts..”(see result section) and that “p53abn tumors had the worst outcomes” (see discussion section). However, they did not demonstrate that. They only reported the median time to progression among the poor responders (11 patients). For the p53abn group (2 patients), the median time to progression or definitive therapy was 9 months (range 9.0-9.0) in the adenocarcinoma cohort and 5 months (range 5.0-5.0) in the EIN cohort; whereas for the p53wt group (6 patients), the median time to progression or definitive therapy was 28.5 months (range 6.0-278 41.0) in the adenocarcinoma cohort and 4 months (range 3.0 – 5.0 months) in the EIN cohort. For the MMRd group (2 patients), it was 8 months (range 8.0-8.0) in the adenocarcinoma group and 36 months (range 280 36.0-36.0) in the EIN group.
  7. Discussion should be re-written. I suggest to structure the discussion as follows: a) start with a summary of the major findings, b) report strengths and limitations of the study, c) compare the paper with the literature, d) add a comment on both interpretation of results and implication of findings in the clinical practice, e) add further future perspectives.
  8. The authors should conclude summarizing the findings and discussion and giving a message to the readers.

Minor points

  1. Method section. The authors classified the endometrial biopsies as “benign complete response”, “atypical - (hormonal effect) partial response, atypical - (improved disease) partial response, atypical - (no better/no worse) stable disease, atypical - progressive disease, or insufficient tissue for evaluation.” Is this classification proposed by the authors? Or is it previously reported or tested by others?
  2. Primary and secondary aims should be reported in the same paragraph one after the other.
  3. Some confusion arises regarding inclusion and exclusion criteria: which are all the eligibility criteria? Have been excluded only patients with their initial endometrial biopsy specimen not available for processing or to have insufficient tissue to undergo the necessary staining for the algorithm?
  4. Strengths and limitations of the study are noy so clearly expressed in the Discussion section; they should be written after the major findings presented in the paragraph.
  5. The authors should conclude giving a message to the readers.

Reviewer 4 Report

In their study, Puechl and co-investigators studied if molecular classification of EIN and early-stage EC treated with LNG-IUS may help to assess the risk for tumor progression. Altogether, 64 patients were selected, but only 58 were enrolled. They were carefully described and all molecular procedures were reported in details. Molecular ProMisE algorithm was applied in all women enrolled.  Eleven out of 58 (19%) progressed or opted for final therapy, and median time was 15.1 months. Interestingly, p53abn tumors /n=4/ revealed the shortest time to progression or definitive therapy. All p53abn tumors  by immunohistochemistry showed TP53 mutations /Table 3/. Finally, the Authors concluded that “ Molecular classification of endometrial cancer or EIN prior to management with LNG-IUS is feasible and may predict patients at risk of progression”.

Altogether, this study is nice, well-prepared and reporting interesting  connection between clinical practice and molecular pathology analysis in a selected group of patients treated with LNG-IUS. Moreover, it was done at Duke University Medical Center, one of the most famous Centers for studying various molecular/immunohistochemical pathways involved in the EC development and progression from many year. I strongly recommend to accept this manuscript for publication, but a minor revision is still necessary.

  1. Please check carefully the manuscript once again and delete spelling error and mistakes for example: progestarone, p53abnl, …as p53 abnormal by abnormal p53 immunohistochemistry…, incombination…
  2. Explain why a patient with EIN received radiation after progression to cancer /?/, not surgical treatment.
  3. 1% of your patients desired future fertility. Please add a few sentences, besides Lynch syndrome counselled, concerning this group of women /genetic testing apart from MLH1 and PMS2, follow-up, recommendations/.
  4. The list of references should be prepared based on the recommendations from the Journal.

Reviewer 5 Report

well written

clinically significant

congratulate authors on job well done

Round 2

Reviewer 2 Report

The authors responded to my comments. The paper sounds.

Reviewer 3 Report

The study has slightly improved after the revision. There are still some points that should be revised.

- Introduction

Line 57 please, delete “most recently”.

Line 59 please specify the % of patients with endometrial cancer and % of patients with EIN who had a relapse.

Line 60 please modify the sentence, I suggest to write “another randomized trial” instead of “a recent..”.

Line 63-63, I suggest to remove this sentence.

The authors report “Previous work has identified clinical factors, which may be associated with more favorable responses, including thickness of the endometrial stripe, younger age at diagnosis, history of prior pregnancy, shorter menstrual cycles, and treatment with megestrol acetate.” However, “thickness of the endometrial stripe” and “treatment with megestrol acetate” are not clinical factors. Moreover, the authors should explicate if “thickness of the endometrial stripe” indicates the thickness at imaging examination or at pathological examination.

Lines 68-74, the sentences are unclear. I suggest first to explicate what it is known and then what it is need to explore.

Primary and secondary aims should be reported one after the other.  Line 102, please remove “therefore”. Please, write the primary objective of the study was…, remove the sentence  “This would identify women at the time of diagnosis with EIN or cancer who may be at increased risk of progression and may not be suitable candidates for medical management” (lines 104-105).

- Methods

Methods are still unclear. The authors did not satisfactory answer to the points raised by the Reviewer.

Inclusion and exclusion criteria are unclear. Lines 118-119, the sentence “Patients who received prior hormonal therapy including, but not limited to, medroxyprogesterone, megestrol acetate, or nore-thindrone prior to or concurrently with LNG-IUS were included” is unclear, please specify. Did the authors include patients with previous hormonal treatments for EIN or endometrial cancer? Had these treatments failures? Or did they include patients who received both LNG-IUS and oral progesterone? This is an important point that should be explicated.

If the authors included patients who had been previously treated with other hormonal treatments, they should explicate, in the result section, how many patients had these treatment and which treatment. Line 120-122 “All patients were counseled regarding the non-standard management and the risk of disease progression.” I do not think that this sentence is necessary, it is a retrospective study and the authors included all patients receiving LNG-IUS.

Line 125 “We identified 58 patients that met these inclusion criteria” this sentence should be removed from the method section.

Primary and secondary end-points should be reported in the same paragraph.

Line 147-148 “Patients could opt for or refuse definitive therapy independent of histologic progression or response.” I suggest to remove this sentence, it is not necessary, as it is a retrospective study.

Line 148-149 “While most providers performed repeat endometrial biopsies every three to six months, these were performed at the discretion of the physician.” I suggest to remove this sentence from the method section and to report the number of biopsies in the result section (i.e. x/X patients had x biopsies performing every three or six months…).

Line 150 “Most patients underwent definitive treatment at the time of progression or lack of response after 12-18 months.” This is a result and it should be removed from the method section.

The authors classified the endometrial biopsies as “benign complete response”, “atypical - (hormonal effect) partial response, atypical - (improved disease) partial response, atypical - (no better/no worse) stable disease, atypical - progressive disease, or insufficient tissue for evaluation.” However, the results have not been shown according to this classification. This point was raised by the Reviewer in the first revision but the authors did not satisfactory answer to this point and they did not change the result section accordingly. I suggest to report in the result section how many patients were classified as benign complete response, how many as atypical - (hormonal effect) partial response, how many as atypical - progressive disease etc…Additionally, I suggest to report these results in all population and in the four different classes. This will make more understandable the results.

- Results section lacks of important findings.

Information on follow up are missing. In particular, I did not find the median follow up and how many biopsies have been performed in the follow up.

I suggest to rewrite the last paragraph in order to underline how many patients had histologic progression in the POLI group, how many in the MMRd etc. This result is shown in the Table but I suggest to report it also in the result section. This is the primary objective of the study and it should be explicated in more details in the result section.

In the method section, the authors stated that imaging findings were reported. However, no information in the result section about imaging has been reported.

- Discussion should be improved

The first paragraph (lines 332-344) should summarize the most important findings reflecting the primary objective of the study: how many POLI patients had histological progression or non-response, how many MMRd patients had progression, how many patients in the p53 wt group had progression etc.. Then the authors should summarize the findings of the secondary objective (see lines  334-336).

Lines 342-343 The authors stated that “p53abn tumors demonstrated the worst outcomes, with the highest proportion of patients requiring definitive therapy (50%) and the shortest time to necessitating definitive therapy (5.7 months).” However, they did not report this in the result section (I could not see any information about time to progress).

Lines 380-389 this paragraph is unclear. I suggest to better explicate the benefit of knowing the molecular classification in patients who desire to preserve fertility and why MMR-D is linked with the Linch Syndrome

In the conclusion paragraph, the authors should report the findings of the primary objective of the study and the implication of findings in the clinical practice.
